# Effect of Cold Drawing and Annealing in Thermomechanical Treatment Route on the Microstructure and Functional Properties of Superelastic Ti-Zr-Nb Alloy

**DOI:** 10.3390/ma16145017

**Published:** 2023-07-15

**Authors:** Anastasia Kudryashova, Konstantin Lukashevich, Mikhail Derkach, Oleg Strakhov, Sergey Dubinskiy, Vladimir Andreev, Sergey Prokoshkin, Vadim Sheremetyev

**Affiliations:** 1Metal Forming Department, National University of Science and Technology MISIS, 119049 Moscow, Russia; kudryashova@misis.ru (A.K.); m142659@edu.misis.ru (K.L.); m144367@edu.misis.ru (M.D.); strakhov.ov@misis.ru (O.S.); prokoshkin@tmo.misis.ru (S.P.); 2A. A. Baikov Institute of Metallurgy and Materials Science of the Russian Academy of Sciences, 119334 Moscow, Russia; andreev.icmateks@gmail.com

**Keywords:** Ti-Zr-Nb shape memory alloy, thermomechanical treatment, drawing, microstructure, crystallographic texture, mechanical properties, functional properties

## Abstract

In this study, a superelastic Ti-18Zr-15Nb (at. %) alloy was subjected to thermomechanical treatment, including cold rotary forging, intermediate annealing, cold drawing, post-deformation annealing, and additional low-temperature aging. As a result of intermediate annealing, two structures of *β*-phase were obtained: a fine-grained structure (d ≈ 3 µm) and a coarse-grained structure (d ≈ 11 µm). Cold drawing promotes grain elongation in the drawing direction; in a fine-grained state, grains form with a size of 4 × 2 µm, and in a coarse-grained state, they grow with a size of 16 × 6 µm. Post-deformation annealing (PDA) at 550 °C for 30 min leads to grain sizes of 5 µm and 3 µm, respectively. After PDA at 550 °C (30 min) in the fine-grained state, the wire exhibits high tensile strength (*UTS* = 624 MPa), highest elongation to failure (*δ* ≥ 8%), and maximum difference between the dislocation and transformation yield stresses, as well as the highest superelastic recovery strain (*ε_r_^SE^* ≥ 3.3%) and total elastic + superelastic recovery strain (*ε_r_^el+SE^* ≥ 5.4%). Additional low-temperature aging at 300 °C for 30–180 min leads to *ω*-phase formation, alloy hardening, embrittlement, and a significant decrease in superelastic recovery strain.

## 1. Introduction

Nickel-free Ti-Zr-Nb-based shape memory alloys (SMAs) are promising materials for medical applications due to their biocompatibility and high level of mechanical and functional properties [1,2,3,4,5,6]. Chemical composition, microstructure, phase state, and crystallographic texture have a significant impact on such properties of these alloys [7,8,9,10]. Considering chemical composition, these alloys can be divided into three groups: low-Zr (≤10 at. % Zr), medium-Zr (10–30 at. % Zr), and high-Zr alloys (≥30 at. % Zr), with maximum theoretical recovery strains of ≤4, 4–6%, and ≥6%, respectively [8]. A single *β*-phase state with a favorable {001}*_β_*<110>*_β_* crystallographic texture provides the highest recovery strains in these alloys [8,11,12]. The most effective tool to control the structural characteristics of these alloys is thermomechanical treatment (TMT) [10,11,12,13,14,15,16,17]. In a number of works [12,17], TMT combining hot/cold rotary forging and post-deformation annealing (PDA), was applied to Ti-(18–19)Zr-(14–15)Nb SMAs with the objective of forming a long-length bar stock 3–8 mm in diameter for bone-implant fabrication. It has been shown that the formation of a polygonized dislocation substructure within *β*-phase grains with an average size of 5–15 μm and a uniform crystallographic texture close to the [001] direction has led to an enhanced combination of the static functional and mechanical properties: relatively high strength (*UTS* = 635 MPa), low Young’s modulus (*E* < 45 GPa), and high superelastic recovery strain (*ε_r_^se^* = 3.4%) [17].

Nowadays, one of the most promising research directions in medical metallurgy and metal science is the miniaturization of implants [18,19,20]. Further miniaturization of medical devices may call for superelastic materials with even smaller dimensions (0.3–3.0 mm in diameter), such as thin wires or thin-wall tubings. Mechanical and functional properties of such materials need to be improved to make it possible to miniaturize them. Drawing, one of the metal forming methods, is the most promising methods for semi-products with a small cross-section (wire). Cold drawing (CD) with a high-quality lubricant and an efficient profile tool makes it possible to obtain a drawn product with a smooth, lucent surface and sufficiently accurate cross-sectional dimensions [21,22].

Drawing is successfully used for the efficient processing of metallic biomaterials based on zinc [23,24], magnesium [25], titanium [26,27], and titanium nickelide [18,28,29,30]. It has been shown that CD of biodegradable Zn- and Mg-based alloys yields control of grain refinement, phase composition, and crystallographic texture to improve strength properties and corrosion behavior [23,24,25]. Cold CD of biomedical *α* + *β*-Ti alloys, due to poor deformability, contributes to change only in phase composition and crystallographic texture [26,27]. However, subsequent PDA can promote grain refinement through static recrystallization. High-temperature deformation, which makes it possible to carry out multi-pass drawings, seems to be more preferable for processing Ti-6Al-4V alloys and promotes grain refinement due to dynamic recrystallization and a corresponding increase in the strength of the alloy [31]. Multi-pass cold drawing, combined with thermal treatment, is effectively used to manufacture thin Ni-Ti wires for vascular stapling and orthodontics [28,29]. The nanocrystalline state with precipitation of dispersed Ti_3_Ni_4_ particles, provided by controlled PDA after CD, makes it possible to achieve abnormally high recovery strains in the thin Ni-Ti wire [32]. In the next work [33], the same authors showed that the strong axial texture <111> formed as a result of CD remains stable after different PDA time–temperature modes, despite recrystallization and an increase in grain size from 37 nm to 9 µm.

To date, there are few works devoted to the study of the impact of TMT, including drawing, on structure and properties of nickel-free Ti-based SMAs [34,35,36,37]. It was shown [35,36] that severe CD of the Ti-29Nb-13Ta-4.6Zr SMA to wires 0.3 mm and 1.0 mm in diameter leads to the formation of a single *β*-phase microstructure, composed of 200 nm thick and 5.0–8.0 μm long grains elongated parallel to the drawing direction. The alloy in this state exhibits high strength (*σ_0.2_* = 410–490 MPa and *UTS* = 740–800 MPa). However, low-ductility (*δ* = 5%) and low-superelastic-recovery strains (*ε_r_^se^* ≈ 0.9%) limit the prospects for application. The possibilities of grain refinement of severally deformed Ti-25Hf-21Nb SMAs using short-term PDA at 650 °C were demonstrated in [34]. Nano- and sub-micro-grained *β*-phase structures formed in thin 127 μm wires as a result of PDA after CD (*e* = 2–3) contribute to an increase in *UTS* up to 850–950 MPa and in *ε_r_^se^* up to 2%. Ductility in this state remains at a low level (*δ* = 4.5%) [34]. Thus, the problem of increasing ductility and recovery strains in Ni-free superelastic wires is relevant. The effect of drawing on microstructure, mechanical, and superelastic properties of Ti-Zr-Nb SMAs remains almost unstudied.

Based on the above data, it can be assumed that drawing will make it possible to form a strongly textured microstructure in the alloy in the drawing direction, which will contribute to the manifestation of an increased set of mechanical and functional properties. Therefore, the idea arose and was implemented in this work to apply combined TMT, including drawing, which is especially effective for obtaining semi-products with a small cross-section. The goal of this work is to study the effect of a new scheme of combined thermomechanical treatment, applied to a Ti-Zr-Nb superelastic alloy including CD on the structure formation, and to establish the relationship between processing regime, structure, and properties.

## 2. Methodology

### 2.1. Investigated Alloy and Its Processing

The object of this study is a Ti-18Zr-15Nb (at. %) SMA, obtained by vacuum arc melting and isostatic pressing (900 °C, 100 MPa, 2 h). After hot isostatic pressing, the ingot was cooled in air and subjected to radial-shear rolling (RSR) at 950 °C and rotary forging (RF) at 700 °C using an *RKM-2* forging machine to obtain 5 mm diameter rods. Then, cold rotary forging (CRF) at RT with an accumulated strain e = 1.5 was carried out. To form a wide range of structural states after cold rotary forging, annealing was carried out at temperatures from 600 to 750 °C (5–120 min) in an air furnace. The evolution of grain structure after all mentioned TMTs is presented in Figure 1, and the measured sizes of the structural elements after annealing are presented in Figure 2.

In the initial state, after hot RF, a homogeneous grain structure is formed, which is a mixture of equiaxed grains with an average size of 30 μm. Subsequent CRF leads to grain elongation in the material extraction direction (ED). PDAs at 600 °C (5–120 min) form a grain size in the range of 3 to 11 µm. After PDA at 700 °C (30–60 min), grain size increases up to 19–32 µm, depending to the annealing time. At an annealing temperature of 750 °C for 30 min, a grain size of ≈31 µm was obtained, and for 60 min, a grain size of ≈38 µm was obtained. In all cases, equiaxed grains are observed and a recrystallized structure is obtained. To assess the effect of initial grain size on the properties of the Ti-Zr-Nb alloy, three different structural states were chosen:600 °C, 5 min (small grain size of ≈3 µm);600 °C, 120 min (intermediate grain size of ≈11 µm);750 °C, 30 min (large grain size of ≈31 µm).

In this work, at first it was necessary to understand how CD affects the Ti-18Zr-15Nb alloy. It was decided to draw the wire until it starts to break. In the fine-grained (FG) state, after PDA at 600 °C (5 min), the wire broke at strain e = 0.8. In the coarse-grained (CG) state, after PDA at 600 °C (120 min), the wire broke at strain e = 1.0. The surface of the rod had the highest degree of defect after PDA at 750 °C (30 min). High-temperature annealing led to the formation of oxide layers, cracks formed on the wire, the die 2.5 mm did not shrink, and so the drawing did not start. Such an effect of RF at 800 °C was previously reported in [12].

The obtained wire was annealed at 550 °C (30 min) in a protective argon atmosphere to form a recrystallized fine-grained structure, which provides an optimal combination of properties [12]. Additional low-temperature aging at 300 °C for 30, 60, and 180 min was applied to the wire to estimate the effect of the *ω*-phase on the mechanical and functional properties of the Ti-Zr-Nb alloy [38,39]. The schedule of thermomechanical treatment is shown in Figure 3.

### 2.2. Experimental Procedure

For microstructure and phase composition study specimens were grinded and polished using a “*SAPHIR 560*”. A longitudinal section of the surface of the specimens from a 10 mm long wire was subjected to grinding on abrasive paper with a suspension of *Eposil F* based on silicon oxide. Ammonia solutions, hydrogen peroxide, and liquid soap were added to the suspension during polishing. Then, the specimens were cleaned in an ultrasonic bath with isopropyl alcohol. The surface was etched in a 1HF:3HNO_3_:6H_2_O solution for 20–60 s.

The grain structure was studied using a “*Versamet-2 Union*” optical microscope. The true average grain size of the *β*-phase was measured using the random linear intercept method [40]. The phase composition was studied by X-ray diffraction analysis using a “*Rigaku Ultima IV*” *(Tokyo, Japan)* diffractometer at RT using *Cu-K_α_* radiation, as well as a parallel beam and graphite monochromator in the 30 to 90 deg 2θ range. For X-ray diffraction analysis, 10 mm long specimens were used in an amount of 6–7 pieces for each process so that when the specimens were located next to each other, the total thickness of the specimens was ≈10 mm.

For microstructure and crystallographic texture investigations, a “*TESCAN VEGA LMH*” scanning electron microscope (SEM) equipped with an electron backscatter diffraction (EBSD) unit, namely the “*NordlysMax2*” detector (Oxford Instruments Advanced AZtecEnergy), was used. Samples were prepared by grinding and mirror-smooth-finish polishing using a “*SAPHIR 560*”. Specimens were tilted by 70° and scanned at 20 kV with a 0.5 µm step.

Static tensile testing to failure was carried out at RT and at a strain rate of 0.02 s^−1^ using an “*Instron 5966*” universal testing machine on 30 mm working length wire specimens. All the measurements were carried out using at least three specimens. From the stress–strain diagrams obtained from static tensile tests, relative elongation to failure (*δ*), yield stress (*σ_0.2_*), transformation yield stress (*σ_tr_*), dislocation yield stress (*σ_dis_*), and ultimate tensile strength *(UTS)* were determined as in [17]. Superelastic cyclic testing was carried out at RT on the same type of specimens according to the scheme “loading the specimen to 1% deformation in the first cycle with a strain increase by 1% in each subsequent cycle for a total strain of 18%”. The superelastic strain recovered upon unloading due to reverse *β→α*″ transformation (*ε_r_^SE^*); the elastic strain recovered upon unloading (*ε^el^*), and their combination (*ε_r_^el+SE^*) was measured from the cyclic stress–strain diagrams, obtained from superelastic cyclic testing. Moreover, according to these diagrams, the accumulated residual strains *ε_acc_*, *σ_tr_*, and *σ_dis_* were determined (Figure 4).

The martensitic transformation start temperature (M_s_) of the Ti-18Zr-15Nb alloy was obtained using the electrical resistivity, measured upon cooling from RT to −150 °C based on the change in the electrical resistivity during the *β→α*″ transformation. The modified ohmmeter–voltmeter method with a four-point connection scheme was used, where ammeter was replaced by voltmeter and a reference resistor of 0.1 Ω and 100 W. The simulations in situ measuring and recording the voltage of the specimen, reference resistor, and K-type thermocouple were performed using a “*PRIST V7-78/1*” millivoltmeter with a 20-channel extension board connected to a PC. A laboratory “*GW Instek SPS-3610*” power supply was used as the current source with *I* = 0.5 A. The specimen working length was of 70 mm with a diameter of 1.5 and 1.7 mm. The samples were cooled in nitrogen vapor at a rate of 5 °C/min using a custom thermal chamber with programmable “*OVEN TRM151*” controller.

## 3. Results and Discussion

### 3.1. Structure–Phase State

The images of the grain structure of the wire after CD and subsequent PDA at a temperature of 550 °C are shown in Figure 5. CD contributes to the elongation of grains in the drawing direction (DD). The average grain size in two dimensions (in the DD and perpendicular to the DD) is 4 × 2 µm after FG+CD and 16 × 6 µm after CG+CD (Figure 5a,b). After PDA at 550 °C for 30 min, the grain size decreases, and the average size is ~5 µm after FG+CD+550, 30 (Figure 5c) and ~3 µm after CG+CD+550, 30 (Figure 5d). In both cases, partial recrystallization occurs, while the dislocation substructure is also partially preserved. Since there was a finer grain structure after FG+CD, and thus accumulated strain energy and grain-boundary energy was higher, the recrystallization process in this case after annealing is faster. Furthermore, after FG+CD+550, 30, a more rapid growth of recrystallized grains occurs, so the grain size in this route is larger [41].

Figure 6a shows X-ray diffractograms of the Ti-Zr-Nb alloy after CRF followed by intermediate annealing, CD, and post-deformation annealing at 550 °C (30 min). As a result of X-ray diffraction analysis, it was revealed that in all cases, the main phase component is the *BCC β*-phase. After CD, some distinct X-ray lines of stress-induced *α*″-martensite are also present. After post-deformation annealing, only *β*-phase X-ray lines are observed. After low-temperature aging at 300 °C for 30, 60, and 180 min, the main phase is also the *β*-phase while it is accompanied by a *ω*-phase (Figure 6b). With an increase in the duration of low-temperature aging, the amount of the *ω*-phase increases too, which is consistent with the results of [42]. Aging for 180 min leads to appearance of a small amount of the *α*-phase.

The crystallographic parameters of the phases detected were calculated as follows. The average value of the BCC *β*-phase lattice parameter after all treatments is the same within the error limits and amounts to *a_β_* = 0.3337 ± 0.0003 nm. The hexagonal *ω*-phase lattice parameter after aging for 60 and 180 min at 300 °C amounts to *a_ω_* = 0.4724 ± 0.0005 nm, *c_ω_* = 0.2886 ± 0.0003 nm, and characteristic ratio *c*/*a* = 0.611 ± 0.007. The orthorhombic lattice parameters of the stress-induced *α*″-phase (martensite) after cold drawing amount to *a_α″_* = 0.3207 ±0.0005 nm, *b_α″_* = 0.5029 ± 0.0006 nm, and *c_α″_* = 0.4685 ± 0.0018.

EBSD images shown in Figure 7a,b demonstrate the microstructure of the wire after PDA at 550 °C (30 min). In Figure 7a,b, the black lines are high-angle boundaries (misorientation angle > 15°). White lines correspond to low-angle boundaries (misorientation angle in the range from 3 to 15°). Grain-size distribution graphs are shown in Figure 7c,d. Analysis of the EBSD images shows that after these TMT routes, a partially polygonised dislocation substructure and partially statically recrystallized structure are formed. After both treatments, large areas of the initially deformed structure are visualized, which are represented as elongated internally polygonized grains containing many low-angle sub-boundaries. Moreover, after CG+CD+550, 30, there are more such deformed areas in which polygonization occurs than after FG+CD+550, 30, which can be explained by the fact that the recrystallization process after FG+CD+550, 30 is faster due to the initially finer grain structure.

Inverse pole figure analysis shows the crystallographic textures of the alloy after FG+CD+550, 30 and CG+CD+550, 30 (Figure 8a,b). For a more complete and clear presentation of the obtained results, these crystallographic textures were supplemented with orientation dependence of the recovery strain limit calculated for the Ti-18Zr-14Nb alloy [43] (Figure 8c). During the *β↔α*″ transformations, the <100>*_β_* and <111>*_β_* orientations correspond to comparatively low theoretical recovery strain limits (~2–3%), while the <110>*_β_* orientation corresponds to a maximum recovery strain limit of 5.7%. The texture with a maximum intensity close to the <102>*_β_* direction parallel to the DD obtained in both cases is of about the same strength. According to Figure 8c, the theoretical limit of the recovery strain can be estimated as ~4.3–5% for these orientations.

### 3.2. Mechanical Properties

Tensile stress–strain diagrams of the Ti-18Zr-15Nb alloy after CRF and subsequent thermomechanical treatments are shown in Figure 9a. The mechanical properties of the Ti-Zr-Nb alloy after different TMT routes, obtained from tensile stress–strain diagrams, are presented in Figure 9b and Table 1.

Static tensile tests to failure showed that after CRF, FG+CD, and CG+CD, a premature failure occurs. Additional CD leads to an increase in tensile strength (*UTS* = 740–750 MPa) and an obvious deterioration in ductility (*δ* = 1%). After FG+CD+550, 30, a high tensile strength (*UTS* = 624 MPa), the highest plasticity (*δ* = 8%), and the maximum difference between the dislocation and transformation yield stress are observed, which later determines the development of irreversible plastic deformation by the dislocation mechanism [12]. Subsequent low-temperature aging at a temperature of 300 °C (30–180 min) showed that with increasing aging time, ultimate tensile strength increases and the alloy becomes less ductile. The aged alloy shows the best combination of ultimate tensile strength (*UTS* ≈ 714 MPa) and ductility (*δ* ≈ 5%) after FG+CD+550, 30+300, 30. After FG+CD+550, 30+300, 60 and FG+CD+550,30+300, 180, specimens were fractured brittlely, which is associated with the development of the precipitation of the embrittling *ω*-phase [39].

### 3.3. Functional Properties

Figure 10 shows cyclic loading–unloading stress–strain diagrams of Ti-Zr-Nb after different TMTs. The superelastic strain recovered due to reverse *β→α*″ transformation (*ε_r_^SE^*), the elastic strain recovered upon unloading (*ε^el^*), and the total recovery strain (*ε_r_^el+SE^*) are among the main quantitative characteristics of the functional behavior of the SMA. These characteristics were measured from the cyclic stress–strain diagrams, obtained from superelastic cyclic testing, as shown in Figure 10.

In fine-grained and coarse-grained states, the alloy exhibits a high value of superelastic strain (3.2% and 2.7%, respectively). After subsequent CD, the alloy in both states does not result in superelastic behavior, and specimens are broken after the second cycle (FG+CD case) and after the third cycle (CG+CD case). Subsequent PDA at 550 °C leads to a significant increase in *ε_r_^el+SE^* values. It should be noted that after PDA of specimens with a fine-grained structure, the alloy exhibits two times greater superelastic recovery strain than in the case of specimens with a coarse-grained structure. After FG+CD+550, 30, the alloy also exhibits the highest superelastic recovery strain (more than 3%) and total recovery strain of 5.4%, which can be explained by the presence of a stronger favorable <102> texture after this treatment. Additional low-temperature aging leads to a decrease in superelastic recovery strain; in all cases, it does not reach 1%. The largest superelastic recovery strain (0.8%) and total recovery strain (3.3%) are maximum after FG+CD+550, 30+300,30, then *ε_r_^el+SE^* decreases with increasing aging time.

Evolutions of the accumulated residual strain, superelastic recovery strain, the transformation, and dislocation yield stresses determined from the cyclic stress–strain diagrams are presented in Figure 11. After all treatments, *ε_r_^SE^* increases; only after FG+CD+550, 30 does it increase to its maximum value before decreasing (Figure 11b). In this latter state, the smallest *ε_acc_* is also observed in the first cycles (Figure 11a). The maximum superelastic recovery strain is observed after FG+CD+550, 30 in the eighth cycle (Figure 11b). The improvement in the superelastic behavior with an increase in the number of cycles occurs due to the difference between the dislocation and transformation yield stresses [17]. With an increase in the number of cycles, a decrease in *σ_tr_* occurs due to the accumulation of oriented internal micro-stresses (Figure 11c) and an increase in the *σ_dis_* due to hardening accumulation (Figure 11d) in accordance with [11]. Apparent critical stress visible during the first two cycles as an only inflection point in the stress–strain diagrams (Figure 11c) cannot be reliably referred to as the transformation or dislocation yield stresses. After CRF, the FG+CD, CG+CD, FG+CD+550, 30+300, 180 alloy exhibits low-ductility (0.5–1%) and low-superelastic-recovery strain (0.1–0.3%). Additional low-temperature aging for 30 and 60 min also leads to low-superelastic-recovery strain: 0.2% and 0.8%, respectively. Thus, based on the results of the mechanical and functional tests, TMTs exhibiting the most favorable combinations of properties can be ranged from the best to worst as follows: FG+CD+550, 30, FG, CG, CG+CD+550, 30.

To explain the above-described differences in the functional behavior of the alloy, electrical resistance measurements were carried out to determine the forward martensitic transformation start temperature M_s_. Figure 12 shows the temperature dependences of the electrical resistivity upon cooling of the Ti-Zr-Nb alloy. Electrical resistivity is represented in an arbitrary unit (*R/R*_0_), where *R*_0_ is the electrical resistivity at 10 °C. The M_s_ temperature of the martensitic transformations minus 112 °C for FG+CD+550, 30 (Figure 12a) is somewhat lower than that of minus 96 °C for CG+CD+550, 30 (Figure 12b), hence the functional properties after the latter TMT should be somewhat worse [1]. However, *ε_r_^SE^* is two times higher in the case of FG+CD+550, 30, while the change in electrical resistivity below M_s_ temperature is almost the same.

### 3.4. General Discussion

To sum up the results of this preliminary study, a comparison of the mechanical properties and the superelastic recovery strain of the Ti-Zr-Nb SMA subjected to conventional high- and low-temperature TMT [12], as well as for nickel-free superelastic alloys subjected to drawing [34,35], is presented in Figure 13. The superelastic recovery strain *ε_r_^SE^* was measured after the total induced strain of 4% for all cases.

The set of mechanical and superelastic properties of the Ti-Zr-Nb bar stock after conventional TMT, including rotary forging, is comparable with the results of this work obtained from wire. However, in this, work low-temperature TMT is applied, the alloy exhibits high recovery strain in the first test cycles comparable to that for the alloy after high-temperature TMT. Given that the alloy after hot deformation exhibits a high functional fatigue life [12,43], such a TMT scheme using drawing looks very promising for future works. Moreover, hot deformation contributes to an increase in the plasticity of the alloy and the ability to be deformed to a greater extent. The ability to apply severe plastic deformation by drawing reveals the likelihood of increasing the strength properties of the wire, along with maintaining excellent superelasticity as was shown for the nickel-free Ti-25Hf-21Nb SMA [34].

Obviously, the potential of drawing has not been exhausted for Ti-Zr-Nb SMAs. This work is primary for further systematic comprehensive research, as well as the study of the mechanisms of the formation and correlation of microstructure, texture, mechanical behavior, and functional characteristics of Ti-Zr-Nb SMAs after drawing.

## 4. Conclusions

A new scheme of combined thermomechanical treatment, including cold drawing, was applied to a Ti-Zr-Nb superelastic alloy. The following relationships between processing regime, structure, and properties were established:Cold drawing of the Ti-18-Zr-15Nb alloy promotes elongated grain direction, with a size of 4 × 2 µm in a fine-grained state and a size of 16 × 6 µm in a coarse-grained state. Subsequent annealing at 550 °C for 30 min leads to grain sizes of 5 µm and 3 µm in these states, respectively. Annealing after drawing in the fine-grained state leads to a larger grain size due to the faster recrystallization process in the initially finer-grained structure;The texture with a maximum intensity in <102>_β_ parallel to the drawing direction obtained after annealing in fine-grained and coarse-grained states is about the same strength, and the theoretical limit of the recovery strain can be estimated as ~4.3–5%;The alloy in a fine-grained state after cold drawing and subsequent post-deformation annealing at 550 °C for 30 min exhibits an excellent combination of the static functional and mechanical properties: relatively high strength (*UTS* = 624 MPa), sufficient ductility (*δ* ≈ 8%), the highest difference between the dislocation and transformation yield stress, and high superelastic recovery strain (*ε_r_^se^_max_* = 3.3%);Additional low-temperature aging at 300 °C for 30–180 min leads to the formation of the *ω*-phase, alloy hardening, a decrease in elongation, and a significant decrease in superelastic recovery strain;The obtained results can be used for the further development of new technologies for manufacturing Ti-Zr-Nb thin wires with an enhanced set of functional properties that provide a high level of biological and biomechanical compatibility with bone tissue.

## Figures and Tables

**Figure 1 materials-16-05017-f001:**
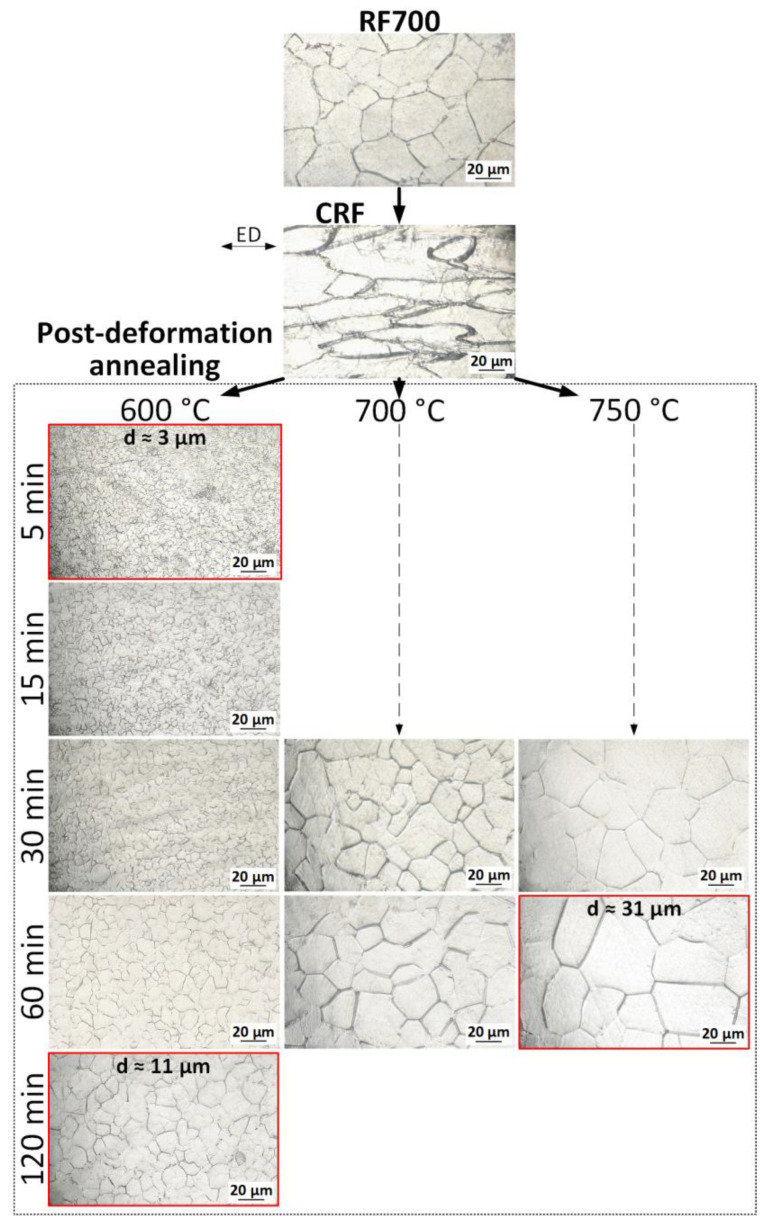
Rod structure after RF700, CRF, and different annealing conditions.

**Figure 2 materials-16-05017-f002:**
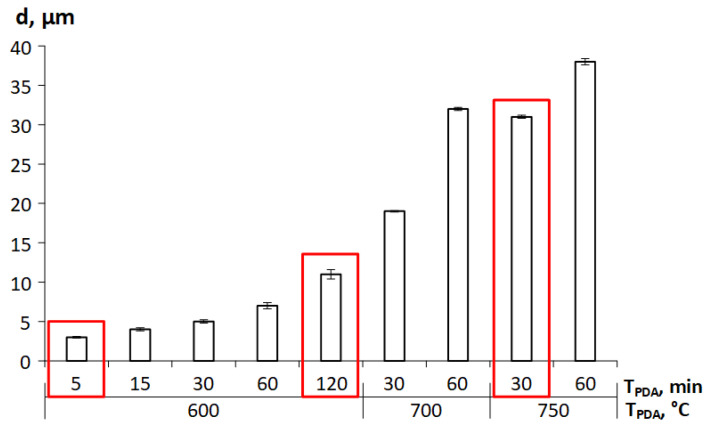
Grain size of Ti-Zr-Nb alloy after different annealing conditions.

**Figure 3 materials-16-05017-f003:**
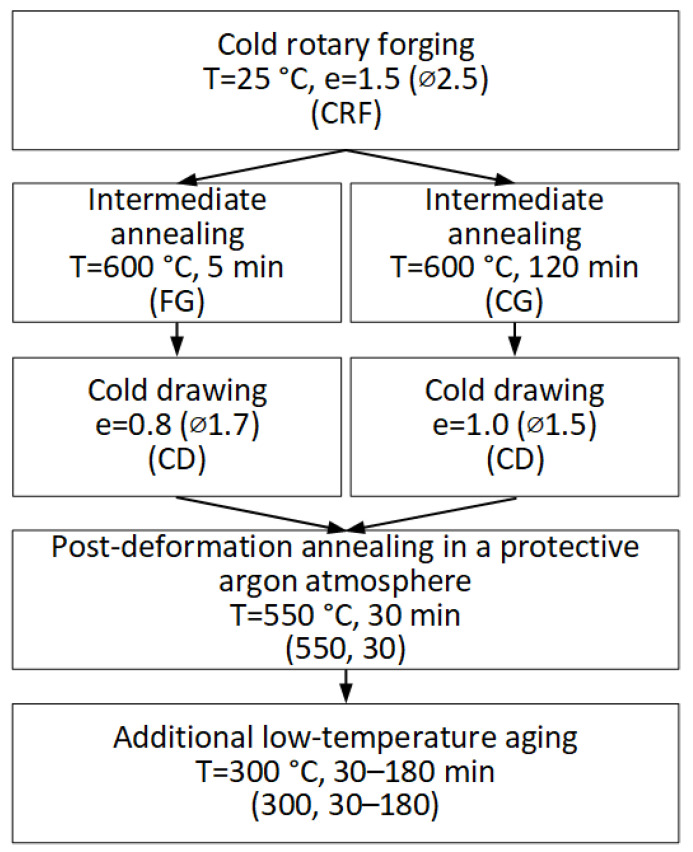
Schedule of thermomechanical treatment.

**Figure 4 materials-16-05017-f004:**
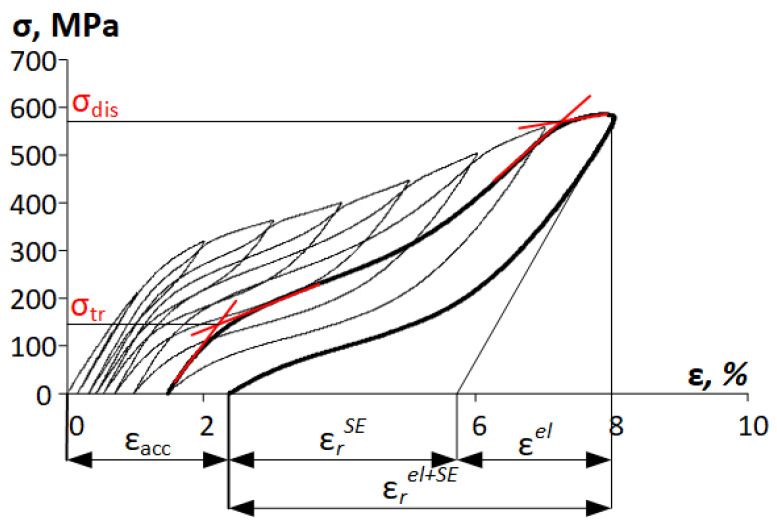
Scheme for determining the elastic strain, superelastic recovery strain, total recovery strain, accumulated residual strain, transformation yield stress, and dislocation yield stress.

**Figure 5 materials-16-05017-f005:**
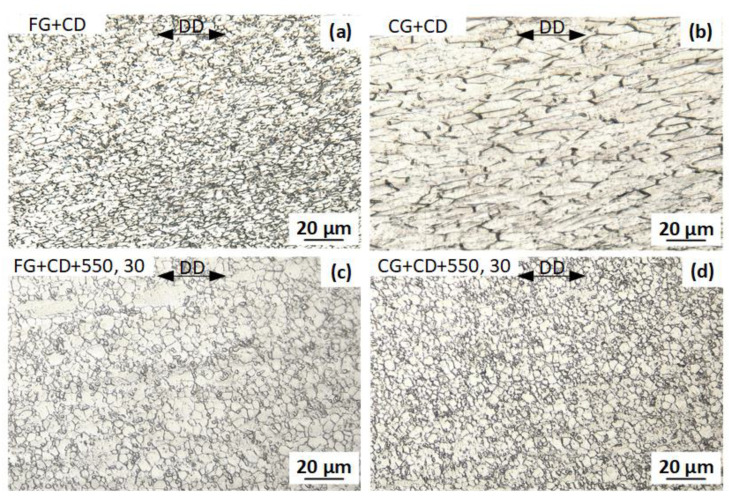
Optical micrographs of wire structure after (**a**) FG+CD; (**b**) CG+CD; (**c**) FG+CD+550, 30; (**d**) CG+CD+550, 30.

**Figure 6 materials-16-05017-f006:**
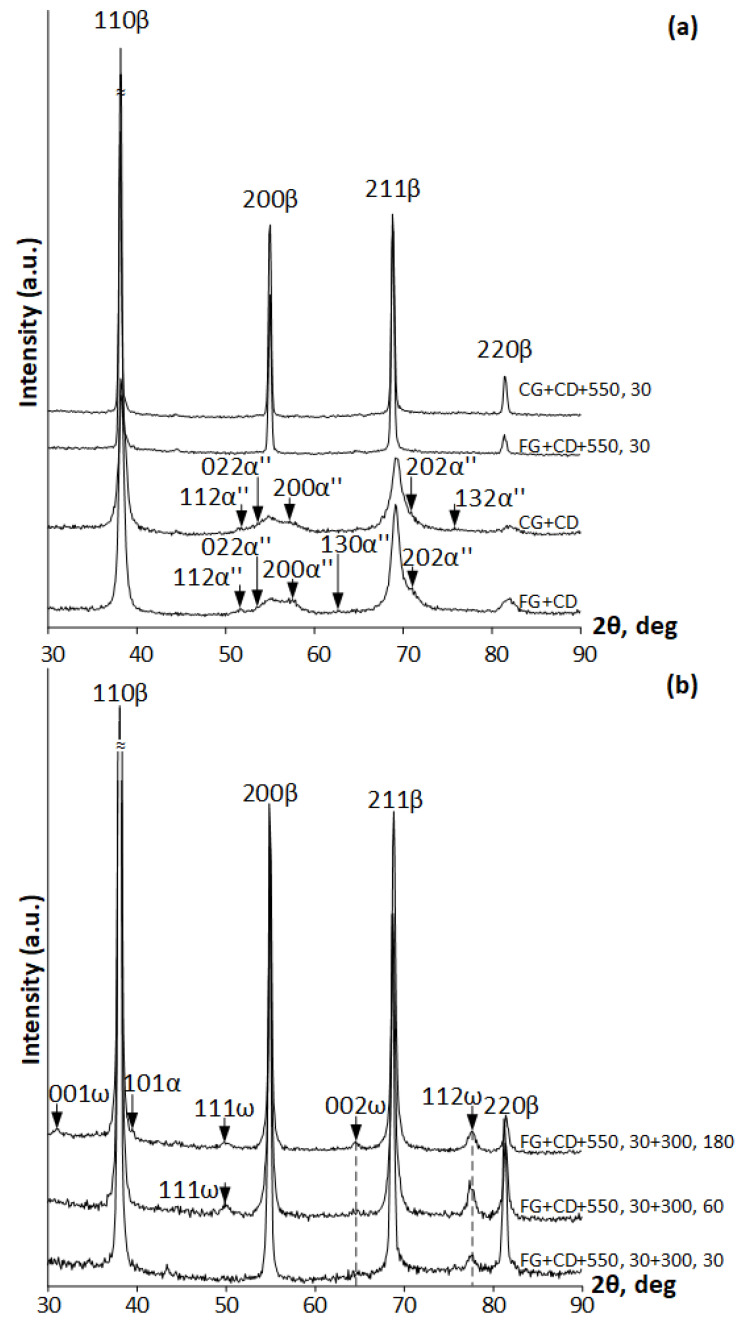
X-ray diffractograms of Ti-Zr-Nb alloy after (**a**) FG+CD, CG+CD, FG+CD+550, 30, CG+CD+550, 30; (**b**) FG+CD+550, 30 and additional low-temperature aging at 300 °C (30–180 min).

**Figure 7 materials-16-05017-f007:**
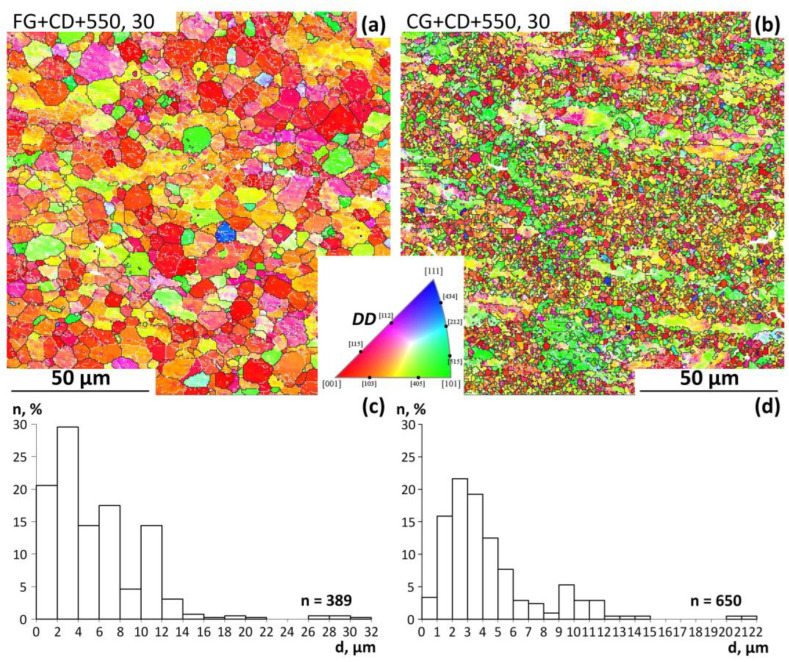
EBSD images after (**a**) FG+CD+ 50, 30 and (**b**) CG+CD+550, 30. DD—drawing direction; grain size distribution after (**c**) FG+CD+550, 30 and (**d**) CG+CD+550, 30.

**Figure 8 materials-16-05017-f008:**
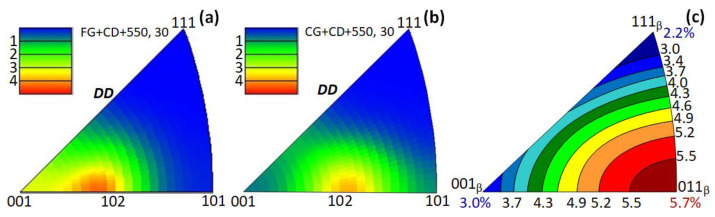
(**a**) Orientation dependence of the recovery strain limit calculated for Ti-18Zr-14Nb alloy [43] and inverse pole figures obtained after: (**b**) FG+CD+550, 30; (**c**) CG+CD+550, 30.

**Figure 9 materials-16-05017-f009:**
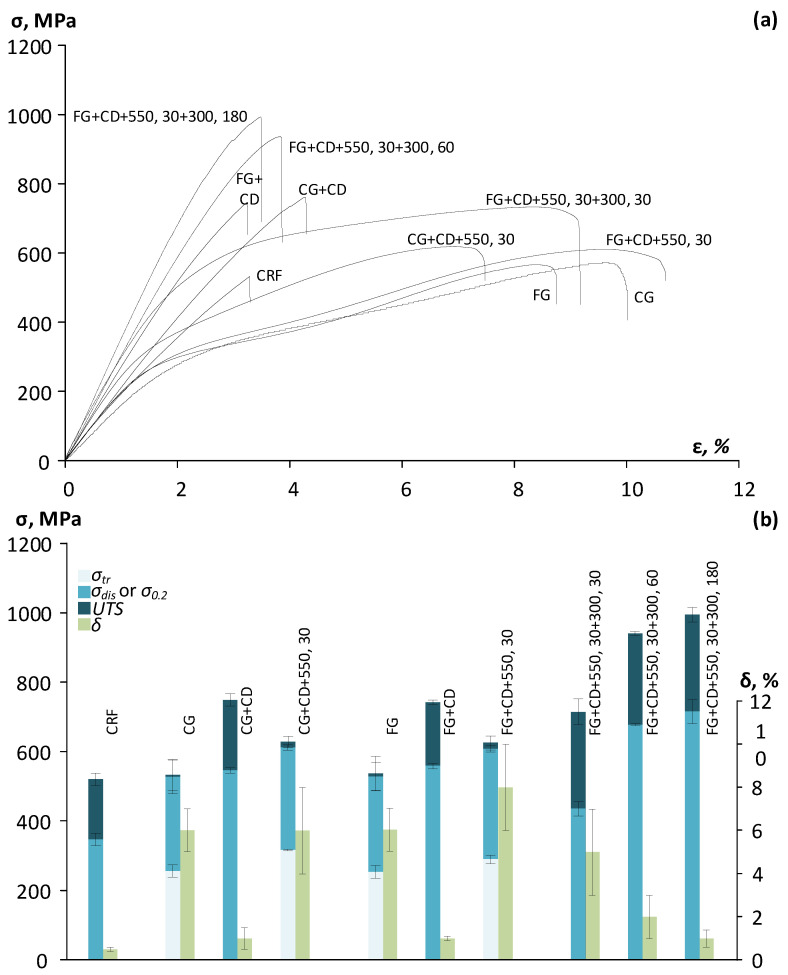
(**a**) Tensile stress–strain diagrams of Ti-Zr-Nb alloy after CRF and following TMT; (**b**) results of the static tensile testing to failure.

**Figure 10 materials-16-05017-f010:**
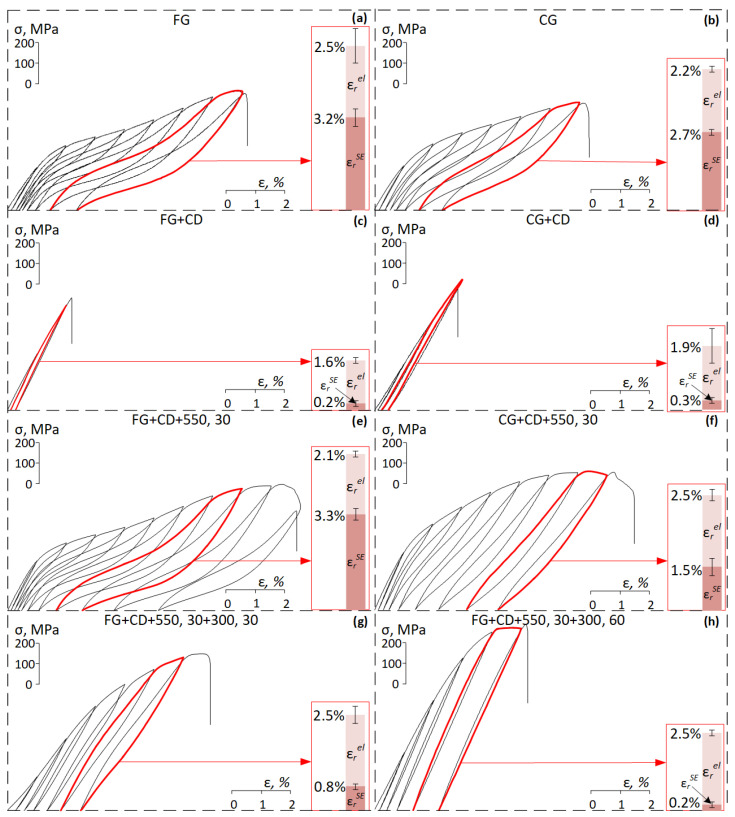
Cyclic stress–strain diagrams of the Ti-Zr-Nb alloy after (**a**) FG; (**b**) CG; (**c**) FG+CD; (**d**) CG+CD; € FG+CD+550, 30; (**f**) CG+CD+550, 30; (**g**) FG+CD+550, 30+300, 30; (**h**) FG+CD+550, 30+300, 60.

**Figure 11 materials-16-05017-f011:**
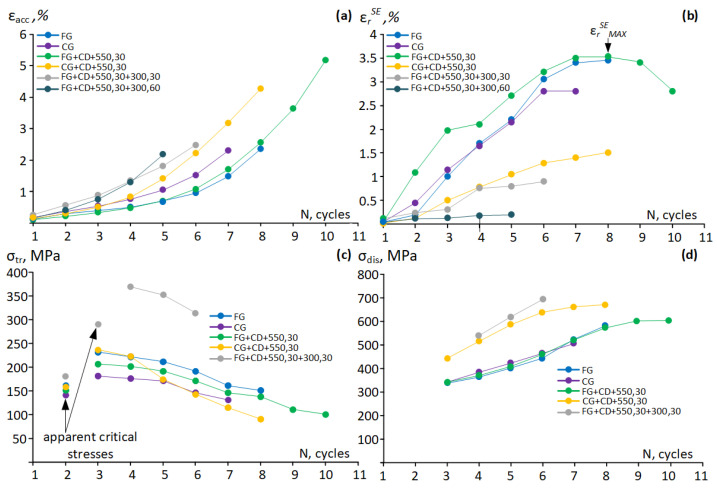
Evolutions of (**a**) accumulated strain; (**b**) superelastic recovery strain; (**c**) transformation yield stress; and (**d**) dislocation yield stress as functions of the number of cycles N.

**Figure 12 materials-16-05017-f012:**
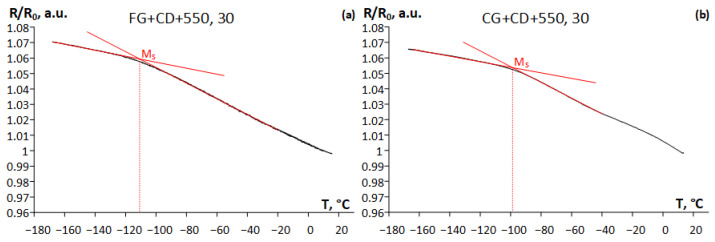
Temperature dependences of the electrical resistance of the Ti-Zr-Nb alloy after (**a**) FG+CD+550, 30 and (**b**) CG+CD+550, 30.

**Figure 13 materials-16-05017-f013:**
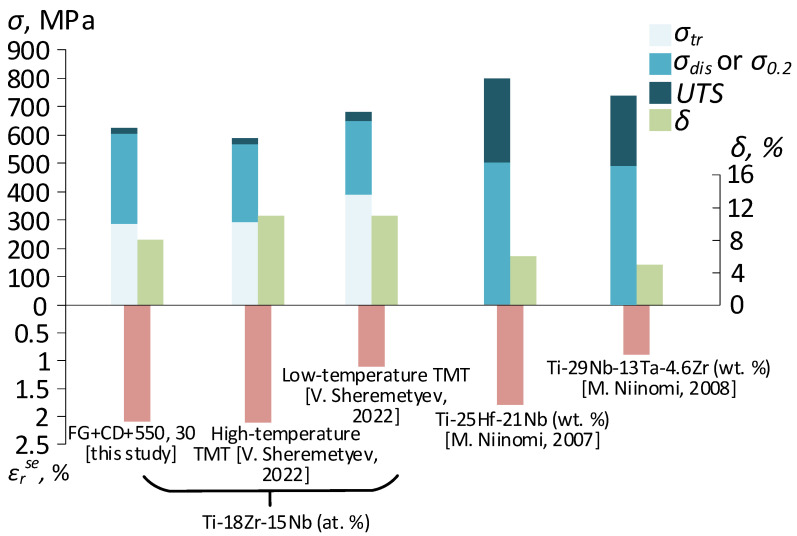
Mechanical and functional properties of the studied alloy and reference alloys [12,35,36].

**Table 1 materials-16-05017-t001:** Mechanical properties of Ti-Zr-Nb alloy after different TMT routes.

TMT Route	*δ*, %	*σ_tr_*	*σ_dis_*, MPa	*UTS*, MPa
CRF	0.5 ± 0.1	-	345 ± 17 *	519 ± 18
CG	6.0 ± 1.0	255 ± 17	526 ± 48	532 ± 45
CG+CD	1.0 ± 0.5	-	544 ± 8 *	748 ± 18
CG+CD+550, 30	6.0 ± 2.0	315 ± 3	611 ± 8	627 ± 16
FG	6.0 ± 1.0	251 ± 18	525 ± 41	535 ± 49
FG+CD	1.0 ± 0.1	-	562 ± 9 *	741 ± 7
FG+CD+550, 30	8.0 ± 2.0	288 ± 12	607 ± 9	624 ± 20
FG+CD+550, 30+300, 30	5.0 ± 2.0	-	434 ± 21 *	714 ± 37
FG+CD+550, 30+300, 60	2.0 ± 1.0	-	675 ± 3 *	940 ± 6
FG+CD+550, 30+300, 180	1.0 ± 0.4	-	714 ± 35 *	994 ± 21

* *σ*_0.2._

## Data Availability

Not applicable.

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
