# Peer review of "Effect of Cold Drawing and Annealing in Thermomechanical Treatment Route on the Microstructure and Functional Properties of Superelastic Ti-Zr-Nb Alloy"

_materials, 2023, doi:10.3390/ma16145017_

Round 1

Reviewer 1 Report

1)In Fig 6a. XRD,  mark is chaos.

2) In Fig.13, check, " Ti-18Zr-15Nb (a t. %) and ...wt.%?

3)Conclusions, should be modifying to clearing.

4) The text should be strengthing to "Mechanical property & XRD &EBSD", 

Reviewer 2 Report

1.       In addition to Figure 7, please include a grain size distribution graph. It appears that CG+CD+550 significantly differs from FG+CD+550. Could you elaborate on the reason behind this difference?

2.       For clarity, please include a table summarizing the key factors (YS, EL, UTS) to complement Figure 9.

3.       From Figure 9, is there any discernible pattern or correlation with heat treatment parameters? The results or curves appear somewhat random.

4.       Could you discuss the correlation between N(cycles) and the stress and strain based on your results? Some insight into this would be beneficial.

Reviewer 3 Report

It is a version of the article that needs a last reading by the authors. For example, on page 15 there is still the title of the patent section.

It is necessary to improve the distribution of the results, highlighting the expected effect of the addition of Nb and Zr in the percentage used.

On the other hand, in the X-ray diffraction analysis, it is necessary to calculate the crystallographic parameters of the different phases detected, as well as the percentage of each phase when they coexist.

No  specific comments, but a general revision is recommended.

Reviewer 4 Report

The authors of the paper : Effect of cold drawing and annealing in thermomechanical treatment route on the microstructure and functional properties of superelastic Ti-Zr-Nb alloy present few experimental results on a interesting field , that of shape memory alloys for medical applications. Few minor corrections can be taken in consideration in order to improve the paper quality: 

L204: mention the microscopies type : optic or SEM 

- mention in section 2 the type of Ebsd, how the samples were prepared for Ebsd and how were the results interpreted 

- in Conclusions section can be inserted a short paragraph about the main results before the four conclusions 

The references section is in accordance with the findings 

English language is acceptable for publication 

Round 2

Reviewer 3 Report

The authors modify the manuscript taking into account the comments of the referees.

No specific comments